# Genetic Analyses of Discrete Geographic Samples of a Golden Chanterelle in Canada Reveal Evidence for Recent Regional Differentiation

**DOI:** 10.3390/genes13071110

**Published:** 2022-06-21

**Authors:** Kuan Zhao, Gregory A. Korfanty, Jianping Xu, R. Greg Thorn

**Affiliations:** 1Department of Biology, McMaster University, Hamilton, ON L8S 4K1, Canada; zhaok35@mcmaster.ca (K.Z.); korfanga@mcmaster.ca (G.A.K.); 2College of Life Science, Jiangxi Science and Technology Normal University, Nanchang 330013, China; 3Department of Biology, University of Western Ontario, London, ON N6A 5B7, Canada

**Keywords:** edible mushroom, microsatellite genotyping, population genetics, range expansion, Hardy–Weinberg equilibrium, excess heterozygotes

## Abstract

The wild edible mushroom *Cantharellus enelensis* is a recently described species of the golden chanterelles found in eastern North America. At present, the genetic diversity and population structure of *C. enelensis* are not known. In this study, we analyzed a total of 230 fruiting bodies of *C. enelensis* that were collected from three regions of Canada: near the east and west coasts of Newfoundland (NFLD), with 110 fruiting bodies each, and around Hamilton, Ontario (10 fruiting bodies). Among the 110 fruiting bodies from each coast in NFLD, 10 from 2009 were without specific site information, while 100 sampled in 2010 were from each of five patches separated by at least 100 m from each other. Each fruiting body was genotyped at three microsatellite loci. Among the total 28 multilocus genotypes (MLGs) identified, 2 were shared among all three regions, 4 were shared between 2 of the 3 regions, and the remaining 22 were each found in only 1 region. Minimal spanning network analyses revealed several region-specific MLG clusters, consistent with geographic specific mutation and expansion. Though the most frequently observed MLGs were shared among local (patch) and regional populations, population genetic analyses revealed that both local and regional geographic separations contributed significantly to the observed genetic variation in the total sample. All three regional populations showed excess heterozygosity; for the eastern NFLD population, we reject the null hypothesis of Hardy–Weinberg equilibrium (HWE) at all three loci. However, the analyses of clone-corrected samples revealed that most loci were in HWE. Together, our results suggest that the three discrete regional populations of *C. enelensis* were likely colonized from a common refugium since the last ice age. However, the local and regional populations are diverging from each other through mutation, drift, and selection at least partly due to heterozygous advantage.

## 1. Introduction

*Cantharellus enelensis* is a popular edible mushroom which is native to eastern North America. This golden chanterelle was historically regarded as part of the *Cantharellus cibarius* species complex until it was described in 2017 [1,2]. The species epithet “*enelensis”* in Latin refers to its presence in NL (the province of Newfoundland and Labrador, Canada), where it is the most common species of chanterelle, both on the island of Newfoundland (NFLD) and in mainland Labrador. However, *C. enelensis* has also been reported in several other parts of eastern North America, from Quebec, Canada to Wisconsin, USA [1,2]. At present, the genetic diversity and population structure of this edible mushroom are unknown. Indeed, little is known about population genetic variations of any chanterelles in eastern North America. The prevalence of *C. enelensis* in NFLD provides an excellent opportunity for studying this group of edible mushrooms.

*Cantharellus enelensis* is an ectomycorrhizal fungus and, consequently, its growth and distribution pattern are closely related to its host plants. The golden chanterelles found in NFLD are associated with *Abies balsamea* (balsam fir), *Picea glauca* (white spruce), and *Picea mariana* (black spruce) [1]. The biogeographic studies of both *Picea* and *Abies* suggested that they originated in western North America and then spread to eastern North America and through the Bering Strait to Eurasia in the early Miocene [3,4]. The last glaciation, which lasted until about 10,000 years ago, covered most of Canada and many northern states in the US [5,6]. The current flora and fauna in these regions were primarily derived from refugium populations south of the glaciation. For *C. enelensis* in NFLD, its sources could be similarly from southern parts of North America or somewhere else, or both. At present, *C. enelensis* has not been reported from outside of eastern North America. Different colonization histories would show different population structures. For example, if there was a single colonization event of NFLD by one individual that subsequently spread across the island, we would see very limited genetic diversity. In contrast, if there were multiple colonization events on the island by different genotypes, genetic differentiations between regional populations may be observed.

In this study, we collected and analyzed a total of 230 fruiting bodies of *C*. *enelensis*. Among these 230, 110 were from five patches at a site near Harbour Grace in eastern NFLD and 110 were from five patches at a site near Stephenville in western NFLD. We also included 10 fruiting bodies from Hamilton, Ontario for comparison. These samples were used to address the following questions: First, how much diversity is there within and between the eastern and western NFLD populations of *C. enelensis*? Second, is there evidence for genetic differentiation between the two regional populations? Third, how does the Hamilton, Ontario population of *C. enelensis* compare with the two NFLD populations, and do they have completely different genotypes? Finally, are there signatures of selection and local adaptation?

To address these questions, we used short tandem repeat (STR) markers developed previously to analyze *Cantharellus formosus*, a close relative of *C. enelensis* [7]. STR markers are among the most frequently used markers to elucidate the relationships among strains and populations of a diversity of organisms, including plants, animals, and fungi [8,9,10,11,12]. Due to their high mutation rate (about 1000 times higher than nucleotide substitutions), these markers are generally highly polymorphic and extremely informative for analyzing recently evolved/expanding populations. For example, Korfanty et al. [11] studied the genetic diversity and dispersal of *Aspergillus fumigatus*, a saprotroph and opportunistic human pathogen, and found that arctic isolates share many alleles with each other and with those collected from other parts of the globe. Similarly, STR markers also helped reveal the genetic diversity and population structure of the wild edible mushroom *Leucocalocybe mongolica* in Mongolian Plateau, East Asia [12]. Here, we screened five STR markers developed for *C. formosus* using a representative set of fruiting bodies of *C. enelensis*. The markers that showed high amplification success and were polymorphic for our samples were used to genotype all 230 fruiting bodies. Aside from addressing the questions above, using the same markers allows us to compare the allelic profiles of the two closely related species, *C. formosus* from western North America and *C. enelensis* from eastern North America.

## 2. Materials and Methods

### 2.1. Sample Collection and DNA Extraction

The samples analyzed here were from three geographic regions. One region is located in eastern NFLD, near Harbour Grace (47°40′48′′ N, 53°12′36″ W, where samples were collected by C. Vilneff, D. Vilneff, and R. Jarvis) and another is in western NFLD, at Sandy Point, near Stephenville (48°27′3″ N, 58°30′51.48″ W, samples collected by C. Vilneff, D. Vilneff, and A. Voitk). The third is in Hamilton, Ontario, Canada (43°17′1″ N, 79°54′32″ W, samples collected by JP Xu). The three discrete collection places are shown in Figure 1, and the geographic map was generated with ArcGIS v10.0 (Esri Canada, Toronto, Canada) [13] http://www.esri.com/software/arcgis/arcgisfor-desktop (accessed on 15 April 2022)). In NFLD, 10 fruiting bodies of *C. enelensis* were sampled from each region on 15 August 2009 and 200 fruiting bodies, 20 from each of 10 local populations, five from eastern NFLD and five from western NFLD, were sampled on 18 and 21 August 2010. Voucher specimens from each site were preserved, as reported previously [1]. Within each site, the fruiting bodies were located at least 50 cm from each other. The geographical distances between local populations (patches) within each of the two regions sampled in 2010 were greater than 100 m. In addition, 10 fruiting bodies were collected from Hamilton, with each located at least 50 cm from each other. Genomic DNA was extracted from each fruiting body using the modified CTAB method [14].

### 2.2. Genotyping

There is currently no genotyping marker for discriminating strains of *C. enelensis*. In addition, the whole-genome sequence of *C. enelensis* is not available to help guide the development of potential polymorphic markers that are specific for this species. Instead, in this study, to identify the genotypes of our specimens, we screened the markers described in the study of *C. formosus* by Dunham et al. [7]. Among the five markers used in that study, one (marker Cf145) failed to amplify any product for our *C. enelensis* samples, even after four attempts using different annealing temperatures. For another marker (Cf339), while we were able to obtain amplification products from our *C. enelensis* samples, DNA sequence analyses of amplified products from six representative specimens of *C. enelensis* (two from each geographic region) revealed no polymorphism within or among these six specimens. However, for the remaining three markers, we successfully obtained amplified products for all specimens and fragment length polymorphisms were observed within and/or among specimens of *C. enelensis*. Those three markers were Cf113, Cf126, and Cf642, and they were respectively labelled using the fluorescent tags HEX, ATTO565, and 6-FAM (IDT). These primers were used to amplify the three STR regions using the following PCR program: pre-denaturation at 94 °C for 3 min followed by 35 cycles of denaturation at 94 °C for 45 s, annealing at 54 °C for 45 s, elongation at 72 °C for 60 s, and a final elongation at 72 °C for 10 min. STR fragments at the three loci were separated using capillary electrophoresis at the MoBix Lab of McMaster University and then analyzed by the STR genotyping software Osiris 2.4 (National Library of Medicine, Bethesda, MD, USA) [15] to determine our fragment length(s) for each marker loci in each specimen.

### 2.3. Allelic and Genotypic Diversities

Allelic diversity including the number of unique alleles (*Na*) and the effective number of alleles (*Ne*) were calculated for each of the three STR loci in each population using the program GenAlEx 6.5 [16,17] that was added to Excel. Unbiased haploid allelic and genotypic diversities (*uh*) were also generated. The unbiased allelic diversity measures the probability that two randomly drawn alleles in the population will be different. Similarly, the unbiased genotypic diversity measures the probability that two randomly drawn individuals will have different multilocus genotypes. Both diversity indices have values that range from 0 to 1. A value of 0 means no variation, where all samples have the same allele at a specific locus or all samples have the same multilocus genotype. In contrast, an *uh* value of 1 means high genetic variation, where any two randomly drawn alleles in the population will be different or any two randomly drawn specimens will have different multilocus genotypes. To determine whether the populations differ significantly in their mean *uh* and mean *Ne*, two-tailed Student’s *t*-tests were conducted.

### 2.4. Relationships among Local and Regional Populations

The program GenAlEx6.5 was used to conduct the analysis of molecular variance (AMOVA) to quantify the contributions of geographic separation to the overall observed genetic variations [18]. Two hierarchical levels of analyses were conducted for the NFLD samples. One level was among local populations (patches) within eastern and western NFLD, and the second level was between the two regional populations.

To visualize the distributions and genetic relationships among multilocus genotypes (MLGs) within and among the local and regional populations of *C. enelensis* in this study, minimum spanning network (MSN) trees were generated using the R package *poppr* [19] based on Bruvo’s genetic distance calculated for each pair of MLGs in the respective samples. Here, the minimum edge genetic distance was set to 0.05. A total of three MSN trees were generated: one for the total population of 230 specimens that included only the three regional populations’ information; the second for the eastern NFLD regional population from 2010 where the five local populations were labeled separately; and the third for the western NFLD population from 2010 where the five local populations were also separately labeled.

### 2.5. STRUCTURE Analysis

The number of potential genetic clusters within our sample was inferred by STRUCTURE v2.3.4 based on an admixture model [20]. Models were tested for *K*-values ranging from 1 to 7, with 10 independent runs per *K* value. For each run, the initial burn-in period was set to 200,000 with 1,200,000 MCMC iterations. To determine the most probable value of *K*, the Δ*K* method was used and implemented in Structure Harvester [21]. In addition, Distruct1.1 [22] and CLUMPP v1.1.2b [23] were used to infer the optimal *K*-cluster affiliations of individual fruiting bodies and geographic populations based on STRUCTURE results, respectively.

### 2.6. Allelic Associations and Mode of Reproduction in Nature

The Hardy–Weinberg equilibrium (HWE) test was conducted to determine whether the observed genotype frequencies at each locus were different from those expected under the hypothesis of random mating and no selection. At present, the mating system of *C. enelensis* is not known. However, the patterns of allelic associations that we observe could provide critical information about its potential mode of reproduction in natural populations. Specifically, in a heterothallic species and in the absence of selection, the genotype frequencies at each locus should approach HWE. In contrast, in homothallic species, we would expect significant deviations of genotype frequencies from HWE, seen as a heterozygote deficit [24]. The GenAlEx 6.5 program was used to determine whether the eastern NFLD, western NFLD, and the Hamilton populations of *C. enelensis* were in HWE [16,17].

### 2.7. Analyses Based on Clone-Corrected Samples

In this study, we collected and analyzed above-ground fruiting bodies of *C. enelensis*. As in other species in the genus *Cantharellus*, *C. enelensis* is an ectomycorrhizal fungus in which most of its biomass is allocated in the soil and tree roots, with fruiting body formation being only a temporary phenomenon induced by environmental cues. In nature, genetic individuals may occupy a range of areas from 5 m^2^ to upwards of 250 m^2^ and produce multiple fruiting bodies [25]. While our mushroom collection method attempted to reduce sampling the same genetic individual (fruiting bodies being 50 cm from each other), in the absence of knowledge about the sizes of genetic individuals, some of the fruiting bodies from the same local site likely belonged to the same genetic individuals. Consequently, some genetic individuals may have been sampled more than once [26]. To reduce the effect of the same genetic individual being sampled multiple times, two data sets were analyzed. The first dataset included all 230 fruiting bodies, and this represented the nonclone-corrected data set. The second dataset included only one MLG per local population and this represented the clone-corrected dataset. Both the HWE tests and the two-level hierarchical AMOVA for the NFLD *C. enelensis* populations were conducted for both the total sample data and the clone-corrected data.

## 3. Results

### 3.1. Genetic Diversity

The STR genotypes were obtained for all 230 samples and analyzed using the methods and programs described above. Our analyses showed that all three markers were polymorphic at all three regional populations, as well as within each of the five local populations in eastern NFLD and in western NFLD. Among the three regional populations, the eastern NFLD regional population had the highest numbers of (i) observed alleles *Na*, (ii) effective alleles *Ne*, and (iii) unbiased haploid genetic diversity *uh*. However, the differences among the three regional populations in each of the three allelic diversity indicators were not statistically significant. Specifically, the p values for the pairwise comparisons in *uh* between eastern NFLD and western NFLD, between eastern NFLD and Hamilton (Ontario), and between western NFLD and Hamilton were 0.36, 0.37, and 0.19, respectively. For the number of observed alleles and effective number of alleles, the differences were also statistically nonsignificant (*p* = 0.64, 0.37, and 0.52 and *p* = 0.39, 0.58, and 0.20, respectively). Together, the results (Table 1) indicated similar levels of allelic diversities among the three discrete regional populations of *C. enelensis*.

### 3.2. Distributions of Multilocus Genotypes

Based on allelic and genotypic information at the three STR loci, we found a total of 28 multilocus genotypes (MLGs) among the 230 fruiting body specimens collected from the two regions in NFLD, as well as the Hamilton, Ontario population. Each region contained multiple MLGs. The distributions of the MLGs within and among the three regions are shown in Figure 1.

Briefly, the numbers of MLGs for the eastern NFLD, western NFLD, and the Hamilton samples were 15, 14, and 7, respectively. The numbers of private MLGs for the 3 regions were 10, 8, and 5, respectively. The frequencies of individual MLGs within each regional population are indicated by the size of the pie diagrams. Among the 28 MLGs detected, 15 were represented by only 1 fruiting body each, with 6 found in the eastern NFLD population, 6 in western NFLD population, and 3 in the Hamilton population. The remaining 13 genotypes were shared by 215 fruiting bodies distributed within and among the populations. Specifically, two MLGs were shared among all three regional populations while four MLGs were shared by two of the three populations. Bruvo’s genetic distances were calculated among the 28 MLGs, and their relationships are shown in Figure 2.

Aside from analyzing the relationships among specimens from the three regions, we also analyzed the samples separately from within the eastern and western NFLD populations to identify the distributions and relationships among local sites within a region. With only 1 exception (Eastern population 1), all local populations (patches) consisted of more than 1 MLG, ranging from 2 to 8 (mean of 4.2) MLGs per patch. There were 11 MLGs in total in the eastern regional population, of which 3 MLGs were represented by only 1 specimen each while 8 were shared by more than 1 specimen within and among the 5 local populations. Of these eight shared MLGs, three were represented by only one local population and the other five were shared by two or more local populations. The most frequently shared MLG was found in 25 specimens from 4 local populations. For the western NFLD regional population, there were 12 MLGs, among which 5 were represented by only 1 specimen each while 7 were shared by more than 1 specimen within and among the 5 local populations, among which 1 MLG was represented by only 1 population and the other 6 MLGs were represented by fruiting bodies from 2 or more local populations. The most frequently shared MLG was found in 22 specimens from all 5 local populations in western NFLD. Together, the results indicated both unique and shared multilocus genotypes among local and regional populations of *C. enelensis* in Canada.

### 3.3. Hardy–Weinberg Equilibrium Test

Comparisons between the observed and expected genotype frequencies revealed differences among the three loci and the three regional populations. In the nonclone-corrected sample, the null hypothesis of HWE was rejected for all three loci in the eastern NFLD population. However, for the western NFLD population, the null hypothesis of HWE was not rejected at two of the three loci. Similarly, for the Hamilton, Ontario population, the genotype frequencies at two of the three loci did not deviate significantly from HWE. After clone-correction, all three loci in the western NFLD population and two of the three loci in eastern NFLD population were in HWE. Locus Cf126 showed a significant excess of heterozygotes for both the Hamilton, Ontario and the eastern NFLD populations, and as a result, we can reject the null hypothesis of HWE (Table 2).

In all six comparisons between observed and expected heterozygosities in the two regional populations in NFLD, the observed heterozygosities were all higher than those expected under random mating in the nonclone-corrected data. In the clone-corrected data, four of the six comparisons had higher observed than expected heterozygosities (Table 2). In the Hamilton population, the same two loci showed greater observed than expected heterozygosities in both the raw and clone-corrected data. However, due to the small sample size in the Hamilton population, the HWE test results here are tentative. Regardless, the results indicate the prevalence of heterozygote excess and the differences among regions in their relative genotype frequencies at the three loci.

### 3.4. STRUCTURE Analyses

The number of potential genetic clusters within the 230 specimens were inferred using STRUCTURE v2.3.4 based on the alleles and genotypes detected at the three marker loci. The admixture model-based simulations indicated that the most suitable ΔK was one, which means the samples collected from the three sites belonged to one inter-breeding metapopulation.

### 3.5. Relationship among Local and Regional Populations in NFLD

Due to the small sample size (10 specimens) of the Hamilton population, we excluded this population from the population genetic differentiation analyses and focused on the two regional populations from NFLD. Specifically, we conducted two-level hierarchical AMOVA analyses for both the nonclone-corrected sample and the clone-corrected sample, excluding the 20 samples collected in 2009. AMOVA results showed that most of the observed genetic variations were found within individual local populations, accounting for 53% and 81% of the total genetic variance for the nonclone-corrected dataset and the clone-corrected dataset, respectively (Table 3).

However, both the regional level and local level separations of the samples also contributed significantly to the observed genetic variation, amounting to 16% and 32%, respectively (*p* < 0.001 in both cases) (Table 3). After clone correction, differences among the local populations decreased from 32% to 0% (making it statistically insignificant), while the regional-level contribution increased from 16% to 19% (Table 3). Regardless, both the nonclone-corrected and clone-corrected data showed statistically significant differences in gene frequencies between the two regional populations of *C. enelensis* in NFLD. We note that even though the two regional populations are statistically differentiated, there are many shared genotypes between them. Specifically, 5 MLGs were shared between the 2 regions, including 64 and 63 specimens from the eastern and western NFLD populations, respectively, accounting for nearly 60% of the total NFLD mushroom samples analyzed in our study. However, both the different frequencies of the shared MLGs and the regional-specific genotypes could have contributed to the observed differentiations (please see also Figure 2). At the local level, the presence of significant differentiation based on the raw data and the lack of it after clone-correction means that the differences among the local populations were due to the over-representation of local site-specific genotypes.

### 3.6. Comparison between the NFLD Population and the Hamilton, Ontario Population

Both similarities and differences were observed between the NFLD population and the Hamilton, Ontario population of *C. enelensis*. For example, the number of unique and effective alleles were similar among the three regional populations: 3 MLGs were shared between the Hamilton and NFLD populations (accounting for a total of 5 and 68 specimens, respectively), and STRUCTURE analyses showed that all 3 regional populations belonged to one inter-breeding metapopulation. However, both unique alleles and unique MLGs were observed in the 2 provinces (NFLD and Ontario), with the NFLD population having 20 MLGs not found in Hamilton while the Hamilton population had 5 MLGs not found in NFLD.

### 3.7. Comparison between the NFLD Populations from 2009 to 2010

We obtained fruiting bodies from two consecutive years (2009 and 2010) from both the east and west coasts of NFLD. Despite the differences in sample sizes between the two years, our comparisons indicated that the observed number of alleles (2.333 ± 0.333 in 2009 vs. 2.033 ± 0.131 in 2010, *n* = 3), the number of effective alleles (1.570 ± 169 in 2009 vs. 1.624 ± 082 in 2010, *n* = 3), and the unbiased gene diversity (0.339 ± 0.083 for 2009 and 0.338 ± 0.039; *n* = 3) were similar between the two years. Together, these results suggested that even a small number of fruiting bodies (20 fruiting bodies with 10 on each coast) were able to capture most of the gene diversity observed in the 200 specimens in NFLD.

At the multilocus genotype level, a total 12 MLGs were found in the 20 fruiting bodies randomly collected in 2009 in NFLD. In 2010, 19 MLGs were found among the 200 fruiting bodies. For the eastern NFLD samples, the number of MLGs increased from 7 in 2009 to 11 in 2010; however, 4 MLGs detected in 2009 were not found in 2010. A similar pattern was observed in western NFLD; the number of MLGs increased from 7 in 2009 to 12 in 2010, but 2 MLGs found in 2009 were not detected in 2010 despite the increased sample sizes. Overall, despite the increased sample sizes and numbers of multilocus genotypes, unbiased genotypic diversities were similar between the two years in both regions in NFLD, as well as in the total NFLD sample. Together, the results are consistent with relatively stable population genetic variations in NFLD with multilocus genotype turnovers, likely due to sexual recombination in both regional populations in NFLD.

## 4. Discussion

In this study, we obtained and genotyped specimens of *C. enelensis* from eastern and western NFLD and from Hamilton, Ontario. While the genetic diversities of other organisms have been reported in various forms in NFLD [9,27,28,29,30], our analysis represents the first attempt to understand the population structure of edible mushrooms on this island. Among the 230 fruiting bodies, and based on genotype information at three STR loci, a total of 28 MLGs were observed. Among these 28, 2 MLGs were shared among all 3 regions while several private MLGs were found from each of the three regions. The allelic diversities were similar to those of *C. formosus* [7] (see below in both the total sample and within each of the three regional populations). Despite the large geographic distances separating them (>2000 km between Hamilton, Ontario and NFLD), because of the proportion of fruiting bodies with shared alleles and the presence of shared MLGs among the three regions, STRUCTURE analysis showed that all 230 samples belonged to 1 genetic cluster. Overall, these results suggest that the three discrete Canadian populations of *C. enelensis* likely shared a common ancestry and/or that there is gene flow among them. However, the statistically significant genetic differentiations among the three regions argue against prevalent gene flow. Indeed, the presence of regional-specific alleles and MLGs that are closely related to each other suggest that the three populations may have been separated from each other for a while. Based on our data, we hypothesize that the three regional populations were colonized from the same refugium population to the south after the retreat of the last glaciation. After colonization, differential mutation, selection, and genetic drift could have contributed to the presence of private alleles and new genotypes within each region.

Indeed, our hypothesis is supported by several lines of evidence. First, among the 3 regions, the 2 shared MLGs account for a total of 49 fruiting bodies, representing 21% of the total sample. Second, among the total 11 alleles obtained here at the 3 loci, 5 were shared among the 3 regions and they represent 45% of the total alleles. Third, the allele- and MLG-sharing was more prevalent between the two regional populations in NFLD, consistent with their geographic proximity, as compared to the Hamilton site. For example, the 2 regions in NFLD shared 5 identical MLGs, corresponding to 64 and 63 samples in each site, accounting for nearly 60% of the total samples from these 2 regions. Fourth, the region-specific alleles were likely due to new mutations accumulated after their colonization at each region. In our samples, there were two private alleles in eastern NFLD, one in western NFLD, and one in Ontario. Fifth, the differential frequencies in shared alleles and shared MLGs could be generated due to genetic drift among the regions. Lastly, evidence for selection is represented by the extent of excess heterozygosities at different loci among the regional populations.

The observation of broad excess heterozygosity within each of the three regional populations of *C. enelensis* was surprising. Most mushroom spores are known to be distributed very close to the fruiting bodies releasing them. Thus, we expected that there would be significant inbreeding within each of the regional populations, resulting in excess homozygosity. The observed excess heterozygosity suggested that the three STR markers themselves, or genes located close to them, may be under selection with heterozygotes being more advantageous than the homozygotes. The different environmental conditions among the regions, including differences in both abiotic and biotic factors, could have contributed to the differences in selection pressures, causing different loci with excess heterozygosity or the same loci with different degrees of (excess) heterozygosity. Indeed, the three regions differed in the mean and ranges of temperatures, soil physical and chemical conditions, humidity, and flora and fauna [27,31,32,33,34,35]. In addition, *C. enelensis* fruiting bodies from eastern NFLD are known to be frequently infested with slugs and maggots (fly larvae), while those in western NFLD are relatively free of slugs and maggots [36]. At present, how these biotic and abiotic factors may influence or be influenced by heterozygosity differences among loci and samples is unknown. Regardless of the potential mechanisms, our observation of excess heterozygosity and potential heterozygous advantage is supported by the results that after clone-correction, only the Cf126 locus in the eastern NFLD and Hamilton populations deviated from HWE, while the remaining seven population–loci combinations were all in HWE.

The frequent finding of HWE in the clone-corrected samples suggests that *C. enelensis* has a heterothallic mating system. Microscopic observations of *C. enelensis* indicated that each basidium typically bears four to eight basidiospores. In basidiomycete mushrooms, basidia with four or more basidiospores each typically can be either homothallic or heterothallic [37,38]. However, in homothallic species, each spore would be self-fertile and expected to contain a single nucleus (or two genetically identical nuclei). Thus, all fruiting bodies derived from such spores would be homozygous and no heterozygotes should be found in nature, which is not what we found here (Table 2). Similarly, the hypothesis of secondary homothallism for *C. enelensis* could also be rejected. In secondarily homothallic species, most basidia typically have two basidiospores on each, with each basidiospore containing two genetically distinct nuclei. Examples of secondary homothallic mushrooms include *Amanita exitialis* [39] and the common cultivars of the commercial button mushroom *Agaricus bisporus* [40,41]. However, our results cannot ascertain whether the proposed heterothallic life cycle in *C. enelensis* is bipolar or tetrapolar. Further research such as whole-genome sequence analyses and crossing experiments using germinated basidiospores are needed to determine whether the mating system in this species is bipolar (unifactorial) or tetrapolar (bifactorial) [42,43,44,45]. Basidiospores of the closely related *C. cibarius* have been germinated 6–12 weeks after being spread on modified Fries medium [43], but no single-spore isolations or matings have been reported.

Because of its historical isolation, the Island of Newfoundland has been the subject of biogeographic and population genetic studies for several species, including humans, other animals, and plants [9,28,29,30,31,32,33,34,35,36,37,38,39,46,47]. For example, a survey of human genetic variation on NFLD revealed evidence for reduced genetic heterogeneity and an increased inbreeding coefficient, compared to that observed in Ireland, Britain, and among Indigenous Peoples of Native America, consistent with inbreeding such as third-cousin marriages as recorded in the literature [30]. In moose (*Alces alces*), the observed low genetic variation in the NFLD population was consistent with founder effects due to the small number of founders that were introduced to the island [9]. In the white pine (*Pinus strobus*), a potential host plant of *C. enelensis*, population genetic analyses revealed little difference between the NFLD population and those in the other regions in eastern Canada, with limited evidence for genetic isolation by geographic distance [28]. Though the number of regions analyzed in our study is smaller than that for the white pine, the lack of a strong signature of genetic isolation by geographic distance among the three *C. enelensis* populations is consistent between these two species. Such a pattern likely reflects their common ancestral populations and, in evolutionary terms, the relatively short time (~10,000 years) that these northern populations of both species have been established since the last glaciation event.

In this study, we used three STR markers to investigate the patterns of genetic variations within and among geographic populations of *C. enelensis*. While the three STR markers showed both unique and similar patterns of variations among the local and regional populations, more markers will likely enable us to identify more MLGs for the 230 analyzed fruiting bodies. However, the small number of MLGs identified here was not entirely due to the lack of discriminating power of our markers. With the current allelic diversity (4, 4, and 3 alleles per locus in the total sample), assuming random mating, the presence of 3 alleles at a locus (i.e., Cf642) could generate 6 possible diploid (or dikaryotic) genotypes and the presence of 4 alleles at a locus (i.e., Cf113 and Cf126) could generate 10 possible diploid genotypes each. If the 3 STR marker loci are not tightly linked with each other, a total of 6 × 10 × 10 = 600 MLGs could be produced from these 3 marker loci. Thus, the large discrepancy between the potential number of genotypes (i.e., 600 MLGs) and the observed number of genotypes (i.e., 28 MLGs) suggest that the overrepresented MLGs in the population were not completely due to lack of power in our markers. Instead, selection and/or genetic linkage have likely played significant roles.

The golden chanterelle in NFLD had been historically labeled as its European counterpart, *C. cibarius*. However, sequence analyses based on both the ITS and the TEF1 sequences revealed its distinctiveness, resulting in it being named a different species (*C. enelensis*) in 2017. We note that though these two gene fragments are commonly used for identifying species of *Cantharellus*, they often contain one- or two-base insertion or deletion (indel) mutations (as well as base substitutions) between haplotype copies within an individual dikaryotic genome, making it difficult to obtain clean sequences from PCR products for these two genes [1]. Often, cloning single copies of the PCR products is needed before sequencing to obtain clean sequences. The frequent indel mutations were also observed among alleles at the three STR markers analyzed here, contributing to their allelic diversities in our populations (Table 4). Similar indel mutations were also detected during STR polymorphism analyses of another chanterelle, *C. formosus*, in western North America [7]. Comparisons of the three loci selected in this study between the two chanterelle species are listed in Table 4. The two species both have three or four alleles at each locus. However, the alleles differ from each other in terms of both the repeat motifs and fragment size ranges. At present, the reasons for the frequent indels in golden chanterelles is not known. Regardless, evidence for the lack of allele-sharing between these two species is consistent with their divergence and reproductive isolation.

## 5. Conclusions and Perspectives

In this study, the genetic diversities and population structures of a gourmet edible mushroom, *C. enelensis*, from three separate areas of Canada were analyzed using three STR loci. Most genetic variations were found within local populations. However, statistically significant genetic differentiations were found among regional populations. Our allelic and genotypic analyses revealed that the three regional populations were likely colonized from the same refugium population after the retreat of the last ice age about 10,000 years ago. The region-specific alleles and genotypes were likely due to new mutations accumulated since their colonization events. Tests for HWE revealed evidence for heterozygote advantage within all three regional populations, but to different degrees. Together, our analyses identified evidence for mutation, drift, and selection in the natural populations of this species.

The golden chanterelle is a popular wild delicacy around the world. In NFLD, aside from the two locations analyzed here, other locations have also reported golden chanterelles, including a restaurant named after this species at the heart of the Gros Morne National Park (https://sugarhillinn.ca/cuisine/, accessed on 1 June 2022), and there is an active commercial harvest of this species [48]. At present, the genetic diversities of *C. enelensis* in other regions in NFLD and how they might be related to these two populations are not known. Similarly, with only 10 fruiting bodies analyzed, the genetic diversity of *C. enelensis* outside of NFLD remains largely unknown. While the current samples allowed us to identify several differences among the local and regional samples, additional samples from other parts of NFLD, as well as from across the broad distribution range in eastern North America, are needed to identify the potential center(s) of diversity for *C. enelensis* and its potential routes of dispersal. Furthermore, only three markers are available to analyze our samples, and this number is lower than those in many recent population genetic studies of fungi. Although these three markers allowed us to reveal several population genetic features of this species, developing and applying additional molecular markers, especially those located in different parts of both the nuclear and mitochondrial genomes, will enhance our ability to discriminate the genotypes among fruiting bodies and provide better assessments of both local and regional population structures. Overall, as an ectomycorrhizal mushroom, its productivity in nature depends on many biotic and abiotic factors, including the host species, nutrients, and physical and chemical properties of the soil [49,50]. To enhance our understanding of this species and to ensure its sustained reproduction and survival in NFLD and other places, additional information about various biotic and abiotic factors associated with local and regional fruiting body samples of *C. enelensis* need to be collected and analyzed to help identify the key factors influencing their reproduction and fruiting in nature.

## Figures and Tables

**Figure 1 genes-13-01110-f001:**
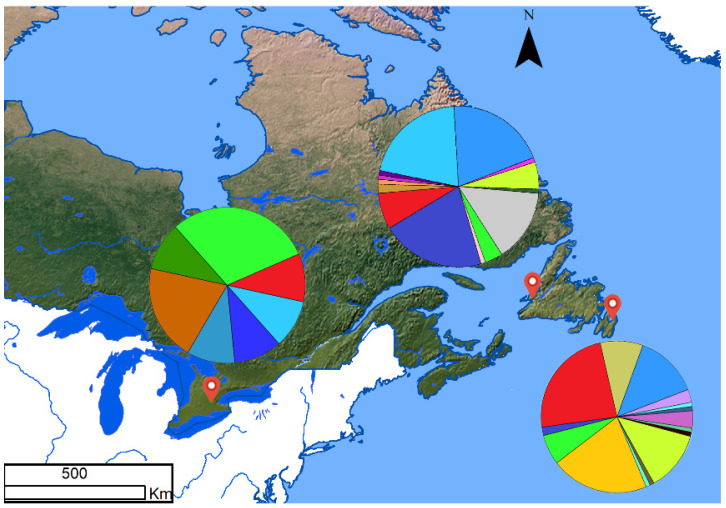
Geographical distribution of 28 MLGs of *C. enelensis*. The frequencies of the MLGs in each site are indicated by the pie diagrams.

**Figure 2 genes-13-01110-f002:**
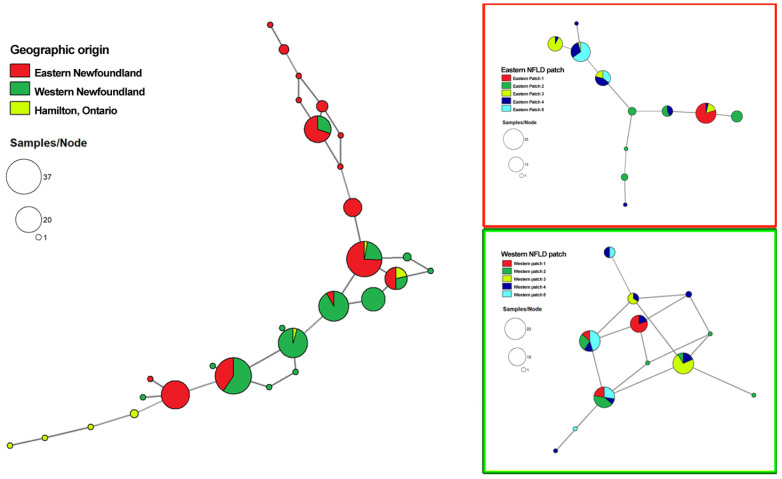
Minimum-spanning network for *C. enelensis* showing the genetic relationship between MLGs. Each circle represents one MLG, where node size corresponds to the number of fruiting bodies for each MLG. The genetic distance between MLGs was calculated using Bruvo’s genetic distance. The MSN for the relationship between all the MLGs from eastern and western NFLD and the mainland (ON) is displayed on the left, whereas the two MSN trees on the right are for the eastern and western NFLD regions.

**Table 1 genes-13-01110-t001:** Number of alleles and allelic diversity at the three microsatellite loci for the three regional populations of *C. enelensis* in Canada.

Population	Locus	*N*	*Na*	*Ne*	*uh*
Eastern NFLD	Cf113	110	4	1.78	0.44
	Cf126	110	3	2.19	0.55
	Cf642	110	2	1.54	0.35
	Mean	110	3	1.84	0.45
	Standard Error	0	0.58	0.19	0.06
Western NFLD	Cf113	110	3	1.96	0.49
	Cf126	110	3	1.55	0.36
	Cf642	110	2	1.07	0.06
	Mean	110	2.67	1.53	0.30
	Standard Error	0	0.33	0.26	0.13
Hamilton, Ontario	Cf113	10	3	2.25	0.58
	Cf126	10	2	1.98	0.52
	Cf642	10	2	1.73	0.44
	Mean	10	2.33	1.98	0.52
	Standard Error	0	0.33	0.15	0.04

*N* = sample size; *Na* = number of unique alleles; *Ne* = effective number of alleles; *uh* = unbiased diversity.

**Table 2 genes-13-01110-t002:** Chi-square tests for Hardy–Weinberg equilibrium within the three regional populations of *C. enelensis* in Canada.

Region	Locus	Nonclone-Corrected			Clone-Corrected		
		Observed Heterozygotes	Expected Heterozygotes	Chi-Square Value	Observed Heterozygotes	Expected Heterozygotes	Chi-Square Value
Eastern NFLD	Cf113	52	20.7	28.504 ***	12	12.8	8.214
	Cf126	103	59.7	85.250 ***	21	13.8	11.931 **
	Cf642	50	38.6	9.516 **	14	10.2	3.529
Western NFLD	Cf113	71	51.0	19.395 ***	18	14.8	2.725
	Cf126	42	38.9	0.984	12	13.4	0.964
	Cf642	7	6.8	0.119	3	2.8	0.086
Hamilton, Ontario	Cf113	8	4.8	4.444	5	1.7	2.160
	Cf126	9	5.4	6.694 **	6	3.4	3.938 *
	Cf642	4	4.2	0.023	3	3.2	0.031

Bold chi-square values indicate that the assuming hypothesis of random mating can be rejected; *p* < 0.05 (*), < 0.01 (**), and < 0.001 (***).

**Table 3 genes-13-01110-t003:** Two-level hierarchical AMOVA of *C. enelensis* in NFLD.

Data	Source	df	SS	MS	Est. Var.	%	Value	*p*
Nonclone-corrected	Among Regions	1	351.580	351.580	2.462	16%	0.160	0.001 ***
	Among Local Pops	8	843.160	105.395	4.865	32%	0.375	0.001 ***
	Within Local Pops	190	1539.000	8.100	8.100	53%	0.475	0.001 ***
	Total	199	2733.740		15.427	100%		
Clone-corrected	Among Regions	1	98.587	98.587	3.505	19%	0.195	0.001 ***
	Among Local Pops	7	93.576	13.368	0.000	0%	−0.018	0.583
	Within Local Pops	40	587.143	14.679	14.679	81%	0.181	0.004 **
	Total	48	779.306		18.183	100%		

Bold *p*-values indicate that the assuming hypothesis of no differentiation can be rejected; *p* < 0.01 (**), and < 0.001 (***).

**Table 4 genes-13-01110-t004:** Comparison of the three microsatellite loci between the two chanterelle species.

Locus	*C. formosus* [7]			*C. enelensis* [This Study]		
	Allele No.	Repeat Motif	Fragment Length *	Allele No.	Repeat Motif	Fragment Length
Cf113	4	(ggt)_5_	96, –, 117, 120	4	(gac)_4_N_38_(acc)_5_	111, 113, 116, 119
Cf126	3	(ggt)_6_	221, 228, 231	4	(ggt)_7_	213, 216, 219, 222
Cf642	4	(gaca)_9_	275, 276, 277, 279	3	(gtct)_4_N_9_(tgtt)_3_	267, 290, 294

*, “–“, allele not reported in paper.

## Data Availability

Not applicable.

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
