# Peer review of "Genetic Analyses of Discrete Geographic Samples of a Golden Chanterelle in Canada Reveal Evidence for Recent Regional Differentiation"

_genes, 2022, doi:10.3390/genes13071110_

Round 1
Reviewer 1 Report
This study by Zhao et al. analyzed the genetic diversities and population structures of C. enelensis, from three separate areas of Canada were using three STR loci. The study is interesting and scientifically sound, but some details or explanations are needed.
Major point:
1. About “ Discrete Population”. In this Ms, Discrete Population seemed as discrete collection places, however, Discrete Population should be discrete population structure or one of the models?
2. Table 2. The data of Cf126 “54”, 54 is correct??
3. I wonder whether there is any influence on the result for the great difference in sample number of each population?
4. And also only 3 STR markers were used, is it enough?
Author Response
Dear Reviewer,
Thank you very much for your review on our manuscript entitled“Genetic analyses of discrete populations of a golden chanterelle in Canada reveal evidence for recent regional differentiation”(Manuscript ID: genes-1758303). We appreciate all your comments and suggestions. We have revised the manuscript accordingly following your suggestions. The changes in the manuscript are all highlighted in the track-changed version attached here.
Below are our responses to your specific comments.
- About “Discrete Population”. In this Ms, Discrete Population seemed as discrete collection places, however, Discrete Population should be discrete population structure or one of the models?
>Response: Thanks for your comment. Indeed, “population” can mean different things to different people. We have revised the title as“Genetic analyses of discrete geographic samples of a golden chanterelle in Canada reveal evidence for recent regional differentiation”. Consequently, the terms have been changed at a few other places where appropriate, including Figure 2.
- Table 2. The data of Cf126 “54”, 54 is correct??
>Response: Thank you for pointing out the mistake. The number should be 5.4. It has been corrected.
- I wonder whether there is any influence on the result for the great difference in sample number of each population?
>Response: Yes, we agree that sample size can have influences on the results. In general, a smaller sample size is less likely to reject the null hypothesis. That is, if a hypothesis (e.g., no genetic differentiation between populations or population being in Hardy-Weinberg equilibrium) is rejected using a small sample size, it’s typically more likely to be rejected if more samples were included. A greater sample size is more likely to detect smaller differences as being statistically significant. In this study, we took a conservation approach and primarily focused on the statistically significant differences due to the rejection of null hypotheses. However, in the revised version, we have added the need for more samples to further substantiate our conclusions and to address additional questions.
- And also only 3 STR markers were used, is it enough?
>Response: Typically, the more markers we use, the more genotypes will likely be detected and there will be more confidence in the conclusions we draw about the analyzed organisms. As stated in the Materials and Methods, we screened five STR markers developed for C. formosus. We intended to use all of them for analyses if they all showed high amplification success and were polymorphic. Unfortunately, only three of them showed polymorphism and were consistently amplified. At present, due to the lack of genome sequence information for this species, screening for additional polymorphic STR markers is not possible. However, as was discussed (Lines 444-458), even with 11 alleles at the three loci, it’s possible to detect 600 multilocus genotypes using these three markers. In the revised version, we have pointed out the need for additional markers to help substantiate our conclusions and to address additional questions.

Reviewer 2 Report
.The study of Thorn and colleagues concerns the genetic diversities and population structure of Cantharellus enelensis from three different areas of Canada. Since C. enelensis is an appreciated edible mushroom, detailed information on the genetic features of its populations will increase the possibility to permit the sustainable collection of this non-timber resource thus ensuring it will continue to carry out its ecological role. The results obtained through this study show that the three populations of C. enelensis differ significantly from a genetic point of view, and that they were likely colonized from a common refugium since the last ice age. In my opinion, the work has been conducted in a complete and appropriate manner, and its findings will be of interest to a vast audience of both mycologists and ecologists.
Author Response
Thank you very much for your review and endorsement of our study.